# Self-supervised Auxiliary Learning
# with Meta-paths for Heterogeneous Graphs

**Dasol Hwang**[1*], **Jinyoung Park**[1*], **Sunyoung Kwon**[4†]
**Kyung-Min Kim**[2,3] , **Jung-Woo Ha**[2,3] , **Hyunwoo J. Kim**[1§]
Korea University[1], NAVER AI LAB[2], NAVER CLOVA[3], Pusan National University[4]
{dd_sol, lpmn678, hyunwoojkim}@korea.ac.kr
skwon@pusan.ac.kr, {kyungmin.kim.ml, jungwoo.ha}@navercorp.com

## Abstract

Graph neural networks have shown superior performance in a wide range of applications providing a powerful representation of graph-structured data. Recent works show that the representation can be further improved by auxiliary tasks. However, the auxiliary tasks for heterogeneous graphs, which contain rich semantic information with various types of nodes and edges, have less explored in the literature. In this paper, to learn graph neural networks on heterogeneous graphs we propose a novel self-supervised auxiliary learning method using meta-paths, which are composite relations of multiple edge types. Our proposed method is learning to learn a primary task by predicting meta-paths as auxiliary tasks. This can be viewed as a type of meta-learning. The proposed method can identify an effective combination of auxiliary tasks and automatically balance them to improve the primary task. Our methods can be applied to any graph neural networks in a plug-in manner without manual labeling or additional data. The experiments demonstrate that the proposed method consistently improves the performance of link prediction and node classification on heterogeneous graphs.

## 1 Introduction

Graph neural networks [1] have been proven effective to learn representations for various tasks such as node classification [2], link prediction [3], and graph classification [4]. The powerful representation yields state-of-the-art performance in a variety of applications including social network analysis [5], citation network analysis [2], visual understanding [6, 7], recommender systems [8], physics [9], and drug discovery [10]. Despite the wide operating range of graph neural networks, employing auxiliary (pre-text) tasks has been less explored for further improving graph representation learning.

Pre-training with an auxiliary task is a common technique for deep neural networks. Indeed, it is the *de facto* standard step in natural language processing and computer vision to learn a powerful backbone networks such as BERT [11] and ResNet [12] leveraging large datasets such as BooksCorpus [13], English Wikipedia, and ImageNet [14]. The models trained on the *auxiliary* task are often beneficial for the *primary* (target) task of interest. Despite the success of pre-training, few approaches have been generalized to graph-structured data due to their fundamental challenges. First, graph structure (e.g., the number of nodes/edges, and diameter) and its meaning can significantly differ between domains. So the model trained on an auxiliary task can harm generalization on the primary task, i.e., *negative transfer* [15]. Also, many graph neural networks are transductive approaches. This often makes transfer learning between datasets inherently infeasible. So, pre-training on the target

dataset has been proposed using auxiliary tasks: graph kernel [16], graph reconstruction [17], and attribute masking [18]. These assume that the auxiliary tasks for pre-training are carefully selected with substantial domain knowledge and expertise in graph characteristics to assist the primary task. Since most graph neural networks operate on *homogeneous* graphs, which have a single type of nodes and edges, the previous pre-training/auxiliary tasks are not specifically designed for *heterogeneous* graphs, which have multiple types of nodes and edges. Heterogeneous graphs commonly occur in real-world applications, for instance, a music dataset has multiple types of nodes (e.g., user, song, artist) and multiple types of relations (e.g., user-artist, song-film, song-instrument).

In this paper, we proposed a framework to train a graph neural networks with automatically selected auxiliary self-supervised tasks which assist the target task without additional data and labels. Our approach first generates meta-paths from heterogeneous graphs without manual labeling and train a model with meta-path prediction to assist the primary task such as link prediction and node classification. This can be formulated as a meta-learning problem. Furthermore, our method can be adopted to existing GNNs in a plug-in manner, enhancing the model performance.

Our **contribution** is threefold: **(i)** We propose a self-supervised learning method on a heterogeneous graph via meta-path prediction without additional data. **(ii)** Our framework automatically selects meta-paths (auxiliary tasks) to assist the primary task via meta-learning. **(iii)** We develop Hint Network that helps the learner network to benefit from challenging auxiliary tasks. To the best of our knowledge, this is the first auxiliary task with meta-paths specifically designed for leveraging heterogeneous graph structure. Our experiment shows that meta-path prediction improves the representational power and the gain can be further improved to explicitly optimize the auxiliary tasks for the primary task via meta-learning and the Hint Network, built on various state-of-the-art GNNs.

## 2  Related Work

**Graph Neural Networks** have provided promising results for various tasks [2, 5–10]. *Bruna et al.* [19] proposed a neural network that performs convolution on the graph domain using the Fourier basis from spectral graph theory. In contrast, non-spectral (spatial) approaches have been developed [2, 20–25]. Inspired by self-supervised learning [26–29] and pre-training [11, 30] in computer vision and natural language processing, pre-training for GNNs has been recently proposed [16, 18]. Recent works show promising results that self-supervised learning can be effective for GNNs [16–18, 31]. Hu et al. [18] have introduced several strategies for pre-training GNNs such as attribute masking and context prediction. Separated from the pre-training and fine-tuning strategy, [31] has studied multi-task learning and analyzed why the pretext tasks are useful for GNNs. However, one problem with both pre-training and multi-task learning strategies is that all the auxiliary tasks are not beneficial for the downstream applications. So, we studied *auxiliary learning* for GNNs that explicitly focuses on the primary task.

**Auxiliary Learning** is a learning strategy to employ auxiliary tasks to assist the primary task. It is similar to multi-task learning, but auxiliary learning cares only the performance of the primary task. A number of auxiliary learning methods are proposed in a wide range of tasks [32–34]. AC-GAN [35] proposed an auxiliary classifier for generative models. Recently, Meta-Auxiliary Learning [36] proposes an elegant solution to generate new auxiliary tasks by collapsing existing classes. However, it cannot be applicable to some tasks such as link prediction which has only one positive class. Our approach generates meta-paths on heterogeneous graphs to make new labels and trains models to predict meta-paths as auxiliary tasks.

**Meta-learning** aims at learning to learn models efficiently and effectively, and generalizes the learning strategy to new tasks. Meta-learning includes black-box methods to approximate gradients without any information about models [37, 38], optimization-based methods to learn an optimal initialization for adapting new tasks [39–41], learning loss functions [40, 42] and metric-learning or non-parametric methods for few-shot learning [43–45]. In contrast to classical learning algorithms that generalize across samples, meta-learning generalizes across tasks. In this paper, we use meta-learning to learn a concept across tasks and transfer the knowledge from auxiliary tasks to the primary task.

# 3 Method

The goal of our framework is to learn with multiple auxiliary tasks to improve the performance of the primary task. In this work, we demonstrate our framework with meta-path predictions as auxiliary tasks. But our framework could be extended to include other auxiliary tasks. The meta-paths capture diverse and meaningful relations between nodes on heterogeneous graphs [46]. However, learning with auxiliary tasks has multiple challenges: identifying useful auxiliary tasks, balancing the auxiliary tasks with the primary task, and converting challenging auxiliary tasks into solvable (and relevant) tasks. To address the challenges, we propose **SEL**f-supervised **A**uxiliary Lea**R**ning (**SELAR**). Our framework consists of two main components: 1) learning weight functions to softly select auxiliary tasks and balance them with the primary task via meta-learning, and 2) learning Hint Networks to convert challenging auxiliary tasks into more relevant and solvable tasks to the primary task learner.

## 3.1 Meta-path Prediction as a self-supervised task

Most existing graph neural networks have been studied focusing on *homogeneous graphs* that have a single type of nodes and edges. However, in real-world applications, *heterogeneous graphs* [47], which have multiple types of nodes and edges, commonly occur. Learning models on the heterogeneous graphs requires different considerations to effectively represent their node and edge heterogeneity.

**Heterogeneous graph** [48]. Let $G = (V, E)$ be a graph with a set of nodes $V$ and edges $E$. A heterogeneous graph is a graph equipped with a node type mapping function $f_v : V \to \mathcal{T}^v$ and an edge type mapping function $f_e : E \to \mathcal{T}^e$, where $\mathcal{T}^v$ is a set of node types and $\mathcal{T}^e$ is a set of edge types. Each node $v_i \in V$ (and edge $e_{ij} \in E$ resp.) has one node type, i.e., $f_v(v_i) \in \mathcal{T}^v$, (and one edge type $f_e(e_{ij}) \in \mathcal{T}^e$ resp.). In this paper, we consider the heterogeneous graphs with $|\mathcal{T}^e| > 1$ or $|\mathcal{T}^v| > 1$. When $|\mathcal{T}^e| = 1$ and $|\mathcal{T}^v| = 1$, it becomes a homogeneous graph.

**Meta-Path** [46, 49] is a path on a heterogeneous graph $G$ that a sequence of nodes connected with heterogeneous edges, i.e., $v_1 \xrightarrow{t_1} v_2 \xrightarrow{t_2} \ldots \xrightarrow{t_l} v_{l+1}$, where $t_l \in \mathcal{T}^e$ denotes an $l$-th edge type of the meta-path. The meta-path can be viewed as a composite relation $R = t_1 \circ t_2 \ldots \circ t_l$ between node $v_1$ and $v_{l+1}$, where $R_1 \circ R_2$ denotes the composition of relation $R_1$ and $R_2$. The definition of meta-path generalizes multi-hop connections and is shown to be useful to analyze heterogeneous graphs. For instance, in Book-Crossing dataset, 'user-item-written.series-item-user' indicates that a meta-path that connects users who like the same book series.

We introduce **meta-path prediction** as a self-supervised auxiliary task to improve the representational power of graph neural networks. To our knowledge, the meta-path prediction has not been studied in the context of self-supervised learning for graph neural networks in the literature.

**Meta-path prediction** is similar to link prediction but meta-paths allow heterogeneous composite relations. The meta-path prediction can be achieved in the same manner as link prediction. If two nodes $u$ and $v$ are connected by a meta-path $p$ with the heterogeneous edges $(t_1, t_2, \ldots t_\ell)$, then $y^p_{u,v} = 1$, otherwise $y^p_{u,v} = 0$. The labels can be generated from a heterogeneous graph without any manual labeling. They can be obtained by $A_p = A_{t_l} \ldots A_{t_2} A_{t_1}$, where $A_t$ is the adjacency matrix of edge type $t$. The binarized value at $(u, v)$ in $A_p$ indicates whether $u$ and $v$ are connected with the meta-path $p$. In this paper, we use meta-path prediction as a self-supervised auxiliary task.

Let $\mathbf{X} \in \mathbf{R}^{|V| \times d}$ and $\mathbf{Z} \in \mathbf{R}^{|V| \times d'}$ be input features and their hidden representations learnt by GNN $f$, i.e., $\mathbf{Z} = f(\mathbf{X}; \mathbf{w}, \mathbf{A})$, where $\mathbf{w}$ is the parameter for $f$, and $\mathbf{A} \in \mathbf{R}^{|V| \times |V|}$ is the adjacency matrix. Then link prediction and meta-path prediction are obtained by a simple operation as

$$\hat{y}^t_{u,v} = \sigma(\Phi_t(z_u)^\top \Phi_t(z_v)), \tag{1}$$

where $\Phi_t$ is the task-specific network for task $t \in \mathcal{T}$ and $z_u$ and $z_v$ are the node embeddings of node $u$ and $v$. e.g., $\Phi_0$ (and $\Phi_1$ resp.) for link prediction (and the first type of meta-path prediction resp.).

**The architecture** is shown in Fig. 1. To optimize the model, as the link prediction, cross entropy is used. The graph neural network $f$ is shared by the link prediction and meta-path predictions. As any auxiliary learning methods, the meta-paths (auxiliary tasks) should be carefully chosen and properly weighted so that the meta-path prediction does not compete with link prediction especially when the capacity of GNNs is limited. To address these issues, we propose our framework that automatically selects meta-paths and balances them with the link prediction via meta-learning.

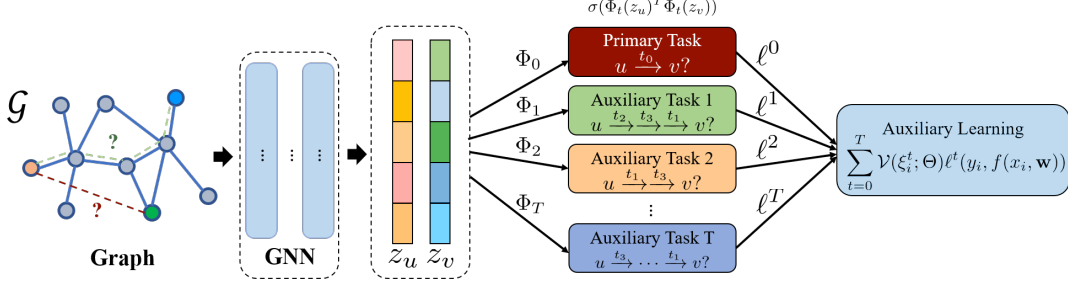

Figure 1: The SELAR framework for self-supervised auxiliary learning. Our framework learns how to balance (or softly select) auxiliary tasks to improve the primary task via meta-learning. In this paper, the primary task is link prediction (or node classification) and auxiliary tasks are meta-path predictions to capture rich information of a heterogeneous graph.

## 3.2 Self-Supervised Auxiliary Learning

Our framework SELAR is learning to learn a primary task with multiple auxiliary tasks to assist the primary task. This can be formally written as

$$\min_{\mathbf{w},\Theta} \mathop{\mathbb{E}}_{(x,y)\sim D^{pr}} \left[ \mathcal{L}^{pr}(\mathbf{w}^*(\Theta)) \right] \quad \text{s.t.} \quad \mathbf{w}^*(\Theta) = \operatorname*{argmin}_{\mathbf{w}} \mathop{\mathbb{E}}_{(x,y)\sim D^{pr+au}} \left[ \mathcal{L}^{pr+au}(\mathbf{w};\Theta) \right] \quad, \quad (2)$$

where $\mathcal{L}^{pr}(\cdot)$ is the primary task loss function to evaluate the trained model $f(x;\mathbf{w}^*(\Theta))$ on meta-data (a validation for meta-learning [40]) $D^{pr}$ and $\mathcal{L}^{pr+au}$ is the loss function to train a model on training data $D^{pr+au}$ with the primary and auxiliary tasks. To avoid cluttered notation, $f$, $x$, and $y$ are omitted. Each task $\mathcal{T}_t$ has $N_t$ samples and $\mathcal{T}_0$ and $\{\mathcal{T}_t\}_{t=1}^T$ denote the primary and auxiliary tasks respectively. The proposed formulation in Eq. (2) learns how to assist the primary task by optimizing $\Theta$ via meta-learning. The nested optimization problem given $\Theta$ is a regular training with properly adjusted loss functions to balance the primary and auxiliary tasks. The formulation can be more specifically written as

$$\min_{\mathbf{w},\Theta} \sum_{i=1}^{M_0} \frac{1}{M_0} \ell^0(y_i^{(0,meta)}, f(x_i^{(0,meta)};\mathbf{w}^*(\Theta))) \tag{3}$$

$$\text{s.t.} \ \mathbf{w}^*(\Theta) = \operatorname*{argmin}_{\mathbf{w}} \sum_{t=0}^{T} \sum_{i=1}^{N_t} \frac{1}{N_t} \mathcal{V}(\xi_i^{(t,train)};\Theta) \ell^t(y_i^{(t,train)}, f^t(x_i^{(t,train)};\mathbf{w})), \tag{4}$$

where $\ell^t$ and $f^t$ denote the loss function and the model for task $t$. We overload $\ell^t$ with its function value, i.e., $\ell^t = \ell^t(y_i^{(t,train)}, f^t(x_i^{(t,train)};\mathbf{w}))$. $\xi_i^{(t,train)}$ is the embedding vector of $i_{th}$ sample for task $t$. It is the concatenation of one-hot representation of task types, the label of the sample (positive/negative), and its loss value, i.e., $\xi_i^{(t,train)} = \left[\ell^t; e_t; y_i^{(t,train)}\right] \in \mathbf{R}^{T+2}$. To derive our learning algorithm, we first shorten the objective function in Eq. (3) and Eq. (4) as $\mathcal{L}^{pr}(\mathbf{w}^*(\Theta))$ and $\mathcal{L}^{pr+au}(\mathbf{w};\Theta)$. This is equivalent to Eq. (2) without expectation. Then, our formulation is given as

$$\min_{\mathbf{w},\Theta} \mathcal{L}^{pr}(\mathbf{w}^*(\Theta)) \quad \text{s.t.} \ \mathbf{w}^*(\Theta) = \operatorname*{argmin}_{\mathbf{w}} \mathcal{L}^{pr+au}(\mathbf{w};\Theta), \tag{5}$$

To circumvent the difficulty of the bi-level optimization, as previous works [39, 40] in meta-learning we approximate it with the updated parameters $\hat{\mathbf{w}}$ using the gradient descent update as

$$\mathbf{w}^*(\Theta) \approx \hat{\mathbf{w}}^k(\Theta^k) = \mathbf{w}^k - \alpha \nabla_{\mathbf{w}} \mathcal{L}^{pr+au}(\mathbf{w}^k;\Theta^k), \tag{6}$$

where $\alpha$ is the learning rate for $\mathbf{w}$. We do not numerically evaluate $\hat{\mathbf{w}}^k(\Theta)$ instead we plug the computational graph of $\hat{\mathbf{w}}^k$ in $\mathcal{L}^{pr}(\mathbf{w}^*(\Theta))$ to optimize $\Theta$. Let $\nabla_{\Theta} \mathcal{L}^{pr}(\mathbf{w}^*(\Theta^k))$ be the gradient evaluated at $\Theta^k$. Then updating parameters $\Theta$ is given as

$$\Theta^{k+1} = \Theta^k - \beta \nabla_{\Theta} \mathcal{L}^{pr}(\hat{\mathbf{w}}^k(\Theta^k)), \tag{7}$$

where $\beta$ is the learning rate for $\Theta$. This update allows softly selecting useful auxiliary tasks (meta-paths) and balance them with the primary task to improve the performance of the primary task.

Without balancing tasks with the weighting function $\mathcal{V}(\cdot; \Theta)$, auxiliary tasks can dominate training and degrade the performance of the primary task.

The model parameters $\mathbf{w}^k$ for tasks can be updated with optimized $\Theta^{k+1}$ in (7) as

$$\mathbf{w}^{k+1} = \mathbf{w}^k - \alpha \nabla_{\mathbf{w}} \mathcal{L}^{pr+au}(\mathbf{w}^k; \Theta^{k+1}). \tag{8}$$

*Remarks.* The proposed formulation can suffer from the meta-overfitting [50, 51] meaning that the parameters $\Theta$ to learn weights for softly selecting meta-paths and balancing the tasks with the primary task can overfit to the small meta-dataset. In our experiment, we found that the overfitting can be alleviated by meta-validation sets [50]. To learn $\Theta$ that is generalizable across meta-training sets, we optimize $\Theta$ across $k$ different meta-datasets like $k$-fold cross validation using the following equation:

$$\Theta^{k+1} = \Theta^k - \beta \mathop{\mathbb{E}}_{D^{pr(meta)} \sim CV} \left[ \nabla_{\Theta} \mathcal{L}^{pr}(\hat{\mathbf{w}}^k(\Theta^k)) \right], \tag{9}$$

where $D^{pr(meta)} \sim CV$ is a meta-dataset from cross validation. We used 3-fold cross validation and the gradients of $\Theta$ w.r.t different meta-datasets are averaged to update $\Theta^k$, see Algorithm 1. The cross validation is crucial to alleviate meta-overfitting and more discussion is Section 4.3.

---

**Algorithm 1** Self-supervised Auxiliary Learning

---

**Input:** training data for primary/auxiliary tasks $D^{pr}, D^{au}$, mini-batch size $N_{pr}, N_{au}$

**Input:** max iterations $K$, # folds for cross validation $C$, learning rate $\alpha, \beta$

**Output:** network parameter $\mathbf{w}^K$ for the primary task

1: Initialize $\mathbf{w}^1, \Theta^1$
2: **for** $k = 1$ to $K$ **do**
3:     $D_m^{pr} \leftarrow \text{MiniBatchSampler}(D^{pr}, N_{pr})$
4:     $D_m^{au} \leftarrow \text{MiniBatchSampler}(D^{au}, N_{au})$
5:     **for** $c = 1$ to $C$ **do**                 ▷ Meta Learning with Cross Validation
6:         $D_m^{pr(train)}, D_m^{pr(meta)} \leftarrow \text{CVSplit}(D_m^{pr}, c)$          ▷ Split Data for CV
7:         $\hat{\mathbf{w}}^k(\Theta^k) \leftarrow \mathbf{w}^k - \alpha \nabla_{\mathbf{w}} \mathcal{L}^{pr+au}(\mathbf{w}^k; \Theta^k)$ with $D_m^{pr(train)} \cup D_m^{au}$    ▷ Eq. (6)
8:         $g_c \leftarrow \nabla_{\Theta} \mathcal{L}^{pr}(\hat{\mathbf{w}}^k(\Theta^k))$ with $D_m^{pr(meta)}$            ▷ Eq. (7)
9:     **end for**
10:     Update $\Theta^{k+1} \leftarrow \Theta^k - \beta \sum_c^C g_c$               ▷ Eq. (9)
11:     $\mathbf{w}^{k+1} = \mathbf{w}^k - \alpha \nabla_{\mathbf{w}} \mathcal{L}^{pr+au}(\mathbf{w}^k; \Theta^{k+1})$ with $D_m^{pr} \cup D_m^{au}$      ▷ Eq. (8)
12: **end for**

---

### 3.3 Hint Networks

Meta-path prediction is generally more challenging than link prediction and node classification since it requires the understanding of long-range relations across heterogeneous nodes. The meta-path prediction gets more difficult when mini-batch training is inevitable due to the size of datasets or models. Within a mini-batch, important nodes and edges for meta-paths are not available. Also, a small learner network, e.g., two-layer GNNs, with a limited receptive field, inherently cannot capture long-range relations. The challenges can hinder representation learning and damage the generalization of the primary task. We proposed a Hint Network (HintNet) which makes the challenge tasks more solvable by correcting the answer with more information at the learner's need. Specifically, in our experiments, the HintNet corrects the answer of the learner with its own answer from the augmented graph with hub nodes, see Fig. 2.

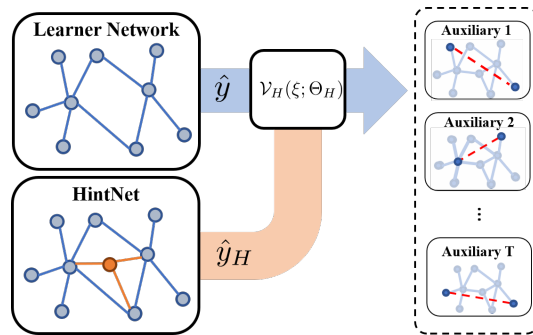

Figure 2: HintNet helps the learner network to learn with challenging and remotely relevant auxiliary tasks. HintNet learns $\mathcal{V}_H$ to decide to use hint $\hat{y}_H$ in the orange line or not. $\hat{y}$ in the blue line denotes the prediction from the learner network.

The amount of help (correction) by HintNet is optimized maximizing the learner's gain. Let $\mathcal{V}_H(\cdot)$ and $\Theta_H$ be a weight function to determine the amount of hint and its parameters which are optimized by meta-learning. Then, our formulation with HintNet is given as

$$\min_{\mathbf{w},\Theta} \sum_{i=1}^{M_0} \frac{1}{M_0} \ell^0(y_i^{(0,meta)}, f(x_i^{(0,meta)}; \mathbf{w}^*(\Theta, \Theta_H))) \tag{10}$$

$$\text{s.t. } \mathbf{w}^*(\Theta) = \underset{\mathbf{w}}{\operatorname{argmin}} \sum_{t=0}^{T} \sum_{i=1}^{N_t} \frac{1}{N_t} \mathcal{V}(\xi_i^{(t,train)}, \ell^t; \Theta) \ell^t(y_i^{(t,train)}, \hat{y}_i^{(t,train)}(\Theta_H)), \tag{11}$$

where $\hat{y}_i^{(t,train)}(\Theta_H)$ denotes the convex combination of the learner's answer and HintNet's answer, i.e., $\mathcal{V}_H(\xi_i^{(t,train)}; \Theta_H) f^t(x_i^{(t,train)}; \mathbf{w}) + (1 - \mathcal{V}_H(\xi_i^{(t,train)}; \Theta_H)) f_H^t(x_i^{(t,train)}; \mathbf{w})$. The sample embedding is $\xi_i^{(t,train)} = \left[ \ell^t; \ell_H^t; e_t; y_i^{(t,train)} \right] \in \mathbf{R}^{T+3}$.

## 4 Experiments

We evaluate our proposed methods on four public benchmark datasets on heterogeneous graphs. Our experiments answer the following research questions: **Q1.** Is meta-path prediction effective for representation learning on heterogeneous graphs? **Q2.** Can the meta-path prediction be further improved by the proposed methods (e.g., SELAR, HintNet)? **Q3.** Why are the proposed methods effective, any relation with hard negative mining?

**Datasets.** We use two public benchmark datasets from different domains for link prediction: Music dataset Last-FM and Book dataset Book-Crossing, released by KGNN-LS [52], RippleNet [53]. We use two datasets for node classification: citation network datasets ACM and Movie dataset IMDB, used by HAN [46] for node classification tasks. ACM has three types nodes (Paper(P), Author(A), Subject(S)), four types of edges (PA, AP, PS, SP) and labels (categories of papers). IMDB contains three types of nodes (Movie (M), Actor (A), Director (D)), four types (MA, AM, MD, DM) of edges and labels (genres of movies). ACM and IMDB have node features, which are bag-of-words of keywords and plots. Dataset details are in the supplement.

**Baselines.** We evaluate our methods with five graph neural networks : GCN [2], GAT [20], GIN [22], SGConv [23] and GTN [24]. Our methods can be applied to both homogeneous graphs and heterogeneous graphs. We compare four learning strategies: **Vanilla**, standard training of base models only with the primary task samples; **w/o meta-path**, learning a primary task with sample weighting function $\mathcal{V}(\xi; \Theta)$; **w/ meta-path**, training with the primary task and auxiliary tasks (meta-path prediction) with a standard loss function; **SELAR** proposed in Section 3.2, learning the primary task with optimized auxiliary tasks by meta-learning; **SELAR+Hint** introduced in Section 3.3. In all the experiments, we report the mean performance of three independent runs. Implementation details are in the supplement. Our experiments were mainly performed based on NAVER Smart Machine Learning platform (NSML) [54, 55].

### 4.1 Learning Link Prediction with meta-path prediction

We used five types of meta-paths of length 2 to 4 for auxiliary tasks. Table 1 shows that our methods consistently improve link prediction performance for all the GNNs, compared to the Vanilla and the method using Meta-Weight-Net [40] only without meta-paths (denoted as w/o meta-path). Overall, a standard training with meta-paths shows 1.1% improvement on average on both Last-FM and Book-Crossing whereas meta-learning that learns sample weights degrades on average on Last-FM and improves only 0.6% on average on Book-Crossing, e.g., GCN, SGC and GTN on Last-FM and GCN and SGC on Book-Crossing, show degradation 0.2% compared to the standard training (Vanilla). As we expected, SELAR and SELAR with HintNet provide more optimized auxiliary learning resulting in 1.9% and 2.0% absolute improvement on Last-FM and 2.6% and 2.7% on the Book-Crossing dataset. Further, in particular, GIN on Book-crossing, SELAR and SELAR+Hint provide ~5.5% and ~5.3% absolute improvement compared to the vanilla algorithm.

Table 1: **Link prediction** performance ($AUC$) of GNNs trained by various learning strategies.

| Dataset | Base GNNs | Vanilla | w/o meta-path | w/ meta-path | Ours SELAR | SELAR+Hint |
|---|---|---|---|---|---|---|
| Last-FM | GCN | 0.7963 | 0.7889 | 0.8235 | **0.8296** | 0.8121 |
| | GAT | 0.8115 | 0.8115 | 0.8263 | 0.8294 | **0.8302** |
| | GIN | 0.8199 | 0.8217 | 0.8242 | **0.8361** | 0.8350 |
| | SGC | 0.7703 | 0.7766 | 0.7718 | 0.7827 | **0.7975** |
| | GTN | 0.7836 | 0.7744 | 0.7865 | 0.7988 | **0.8067** |
| | Avg. Gain | - | -0.0017 | +0.0106 | +0.0190 | +0.0200 |
| Book-Crossing | GCN | 0.7039 | 0.7031 | 0.7110 | 0.7182 | **0.7208** |
| | GAT | 0.6891 | 0.6968 | 0.7075 | 0.7345 | **0.7360** |
| | GIN | 0.6979 | 0.7210 | 0.7338 | **0.7526** | 0.7513 |
| | SGC | 0.6860 | 0.6808 | 0.6792 | 0.6902 | **0.6926** |
| | GTN | 0.6732 | 0.6758 | 0.6724 | **0.6858** | 0.6850 |
| | Avg. Gain | - | +0.0055 | +0.0108 | +0.0263 | +0.0267 |

Table 2: **Node classification** performance ($F1$-score) of GNNs trained by various learning schemes.

| Dataset | Base GNNs | Vanilla | w/o meta-path | w/ meta-path | Ours SELAR | SELAR+Hint |
|---|---|---|---|---|---|---|
| ACM | GCN | 0.9091 | 0.9191 | 0.9104 | 0.9229 | **0.9246** |
| | GAT | 0.9161 | 0.9119 | 0.9262 | 0.9273 | **0.9278** |
| | GIN | 0.9085 | 0.9118 | 0.9058 | 0.9092 | **0.9135** |
| | SGC | 0.9163 | 0.9194 | 0.9223 | 0.9224 | **0.9235** |
| | GTN | 0.9181 | 0.9191 | 0.9246 | **0.9258** | 0.9236 |
| | Avg. Gain | - | +0.0027 | +0.0043 | +0.0079 | **+0.0090** |
| IMDB | GCN | 0.5767 | 0.5855 | 0.5994 | 0.6083 | **0.6154** |
| | GAT | 0.5653 | 0.5488 | 0.5910 | **0.6099** | 0.6044 |
| | GIN | 0.5888 | 0.5698 | 0.5891 | **0.5931** | 0.5897 |
| | SGC | 0.5779 | 0.5924 | 0.5940 | 0.6151 | **0.6192** |
| | GTN | 0.5804 | 0.5792 | 0.5818 | 0.5994 | **0.6063** |
| | Avg. Gain | - | -0.0027 | +0.0132 | +0.0274 | **+0.0292** |

## 4.2 Learning Node Classification with meta-path prediction

Similar to link prediction above, our SELAR consistently enhances node classification performance of all the GNN models and the improvements are more significant on IMDB which is larger than the ACM dataset. We believe that ACM dataset is already saturated and the room for improvement is limited. However, our methods still show small yet consistent improvement over all the architecture on ACM. We conjecture that the efficacy of our proposed methods differs depending on graph structures. However, it is worth noting that introducing meta-path prediction as auxiliary tasks remarkably improves the performance of primary tasks such as link and node prediction with consistency compared to the existing methods. "w/o meta-path", the meta-learning to learn sample weight function on a primary task shows marginal degradation in five out of eight settings. Remarkably, SELAR improved the F1-score of GAT on the IMDB by (4.46%) compared to the vanilla learning scheme.

## 4.3 Analysis of Weighting Function and Meta-overfitting

The effectiveness of meta-path prediction and the proposed learning strategies are answered above. To address the last research question **Q3.** why the proposed method is effective, we provide analysis on the weighting function $\mathcal{V}(\xi; \Theta)$ learned by our framework. Also, we show the evidence that meta-overfitting occurs and can be addressed by cross-validation as in Algorithm 1.

**Weighting function.** Our proposed methods can automatically balance multiple auxiliary tasks to improve the primary task. To understand the ability of our method, we analyze the weighting function

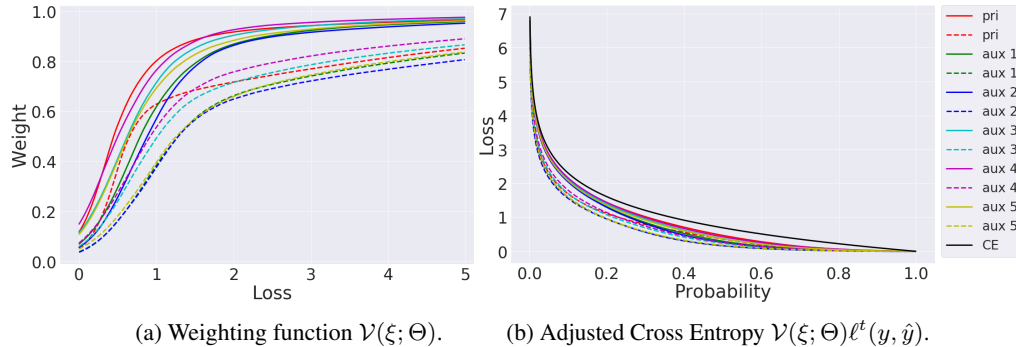

(a) Weighting function $\mathcal{V}(\xi; \Theta)$.    (b) Adjusted Cross Entropy $\mathcal{V}(\xi; \Theta)\ell^t(y, \hat{y})$.

Figure 3: Weighting function $\mathcal{V}(\cdot)$ learnt by SELAR+HintNet. $\mathcal{V}(\cdot)$ gives overall high weights to the primary task positive samples (red) in (a). $\mathcal{V}(\cdot)$ decreases the weights of easy samples with a loss ranged from 0 to 1. In (b), the adjusted cross entropy, i.e., $-\mathcal{V}(\xi; \Theta)\log(\hat{y})$, by $\mathcal{V}(\cdot)$ acts like the focal loss, which focuses on hard examples by $-(1 - p_t)^\gamma \log(\hat{y})$.

and the adjusted loss function by the weighting function, i.e.,$\mathcal{V}(\xi; \Theta)$, $\mathcal{V}(\xi; \Theta)\ell^t(y, \hat{y})$. The positive and negative samples are solid and dash lines respectively. We present the weighting function learnt by SELAR+HintNet for GAT which is the best-performing construction on Last-FM. The weighting function is from the epoch with the best validation performance. Fig. 3 shows that the learnt weighting function attends to hard examples more than easy ones with a small loss range from 0 to 1.

Also, the primary task-positive samples are relatively less down weighted than auxiliary tasks even when the samples are easy (i.e., the loss is ranged from 0 to 1). Our adjusted loss $\mathcal{V}(\xi; \Theta)\ell^t(y, \hat{y})$ is closely related to the focal loss, $-(1 - p_t)^\gamma \log(p_t)$. When $\ell^t$ is the cross-entropy, it becomes $\mathcal{V}(\xi; \Theta)\log(p_t)$, where $p$ is the model's prediction for the correct class and $p_t$ is defined as $p$ if $y = 1$, otherwise $1 - p$ as [56]. The weighting function differentially evolves over iterations. At the early stage of training, it often focuses on easy examples first and then changes its focus over time. Also, the adjusted loss values by the weighting function learnt by our method differ across tasks. To analyze the contribution of each task, we calculate the average of the task-specific weighted loss on the Last-FM and Book-Crossing datasets. Especially, on the Book-Crossing, our method has more attention to 'user-item' (primary task) and 'user-item-literary.series.item-user' (auxiliary task) which is a meta-path that connects users who like a book series. This implies that two users who like a book series likely have a similar preference. More results and discussion are available in the supplement.

**Meta cross-validation**, i.e., cross-validation for meta-learning, helps to keep weighting function from over-fitting on meta data. Table 3 evidence that our algorithms as other meta-learning methods can overfit to meta-data. As in Algorithm 1, our proposed methods, both SELAR and SELAR with HintNet, with cross-validation denoted as '3-fold' alleviates the meta-overfitting problem and provides a significant performance gain, whereas without meta cross-validation denoted as '1-fold' the proposed method can underperform the vanilla training strategy.

Table 3: Comparison between 1-fold and 3-fold as meta-data on **Last-FM** datasets.

| Model | Vanilla | SELAR 1-fold | SELAR 3-fold | SELAR+Hint 1-fold | SELAR+Hint 3-fold |
|---|---|---|---|---|---|
| GCN | 0.7963 | 0.7885 | **0.8296** | 0.7834 | **0.8121** |
| GAT | 0.8115 | 0.8287 | **0.8294** | 0.8290 | **0.8302** |
| GIN | 0.8199 | 0.8234 | **0.8361** | 0.8244 | **0.8350** |
| SGC | 0.7703 | 0.7691 | **0.7827** | 0.7702 | **0.7975** |
| GTN | 0.7836 | 0.7897 | **0.7988** | 0.7915 | **0.8067** |

## 5    Conclusion

We proposed meta-path prediction as self-supervised auxiliary tasks on heterogeneous graphs. Our experiments show that the representation learning on heterogeneous graphs can benefit from meta-

path prediction which encourages to capture rich semantic information. The auxiliary tasks can be further improved by our proposed method SELAR, which automatically balances auxiliary tasks to assist the primary task via a form of meta-learning. The learnt weighting function identifies more beneficial meta-paths for the primary tasks. Within a task, the weighting function can adjust the cross entropy like the focal loss, which focuses on hard examples by decreasing weights for easy samples. Moreover, when it comes to challenging and remotely relevant auxiliary tasks, our HintNet helps the learner by correcting the learner's answer dynamically and further improves the gain from auxiliary tasks. Our framework based on meta-learning provides learning strategies to balance primary task and auxiliary tasks, and easy/hard (and positive/negative) samples. Interesting future directions include applying our framework to other domains and various auxiliary tasks. Our code is publicly available at `https://github.com/mlvlab/SELAR`.

**Acknowledgements.** This work was partly supported by NAVER Corp. and Institute for Information & communications Technology Planning & Evaluation (IITP) grants funded by the Korea government (MSIT): the Regional Strategic Industry Convergence Security Core Talent Training Business (No.2019-0-01343) and the ICT Creative Consilience Program (IITP-2020-0-01819).

## Broader Impact

We thank NeurIPS2020 for this opportunity to revisit the broader impact of our work and the potential societal consequence of machine learning researches. Our work is a general learning method to benefit from auxiliary tasks. One interesting finding is that meta-path prediction can be an effective self-supervised task to learn more power representation of heterogeneous graphs. Nowadays, people use social media (e.g., Facebook, Twitter, etc.) on a daily basis. Also, people watch movies and TV-shows online and purchase products on Amazon. All this information can be represented as heterogeneous graphs. We believe that our meta-path auxiliary tasks will benefit the customers with improved services. For instance, more accurate recommender systems will save customers' time and provide more relevant contents and products. We believe that there is no direct negative consequence of this research. We proposed how to train models with auxiliary tasks. We did not make any algorithms for specific applications. So, no one will be put at a disadvantage from our work. No direct negative consequence of a failure of the system is expected. We used four datasets Last-FM, Book-Crossing, ACM, and IMDB. They may not represent all the population on the earth but our experiments did not leverage any biases in the datasets. We believe that our method will be as effective as we reported in the paper on different datasets from different populations.

## Footnotes

*First two authors have equal contribution. § is the corresponding author.

†This work was done when the author worked at NAVER CLOVA.

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
