[Supplementary Material]

# Self-supervised Auxiliary Learning
# with Meta-paths for Heterogeneous Graphs
# (Supplement)

**Dasol Hwang**[1*], **Jinyoung Park**[1*], **Sunyoung Kwon**[4†]
**Kyung-Min Kim**[2,3] , **Jung-Woo Ha**[2,3] , **Hyunwoo J. Kim**[1]
Korea University[1], NAVER AI LAB[2], NAVER CLOVA[3], Pusan National University[4]
{dd_sol, lpmn678, hyunwoojkim}@korea.ac.kr
skwon@pusan.ac.kr, {kyungmin.kim.ml, jungwoo.ha}@navercorp.com

## A  Summary

We provide additional results and implementation details that are not included in the main paper due to the space limit. This supplement includes (1) details of datasets, (2) implementation details, (3) task selection results, and (4) behaviors of the weighting function at different training stages.

## B  Details of datasets

We use two datasets (Last-FM, Book-Crossing) for link prediction tasks and two datasets (ACM, IMDB) for node classification tasks. Last-FM and Book-Crossing do not have node features, while ACM and IMDB have node features, which are bag-of-words of keywords and plots. The Last-FM dataset with a knowledge graph has 122 types of edges, e.g., "artist.origin", "musician.instruments.played", "person.or.entity.appearing.in.film", and "film.actor.film", etc. Book-Crossing with a knowledge graph has 52 types of edges, e.g., "book.genre", "literary.series", "date.of.first.publication", and "written.work.translation", etc. ACM has three types of nodes (Paper(P), Author(A), Subject(S)), four types of edges (PA, AP, PS, SP), and labels (categories of papers). IMDB contains three types of nodes (Movie (M), Actor (A), Director (D)), four types (MA, AM, MD, DM) of edges and labels (genres of movies). Statistics of the datasets are in Table 1.

Table 1: Datasets on heterogeneous graphs.

|  | Datasets | # Nodes | # Edges | # Edge type | # Features |
|---|---|---|---|---|---|
| Link prediction | Last-FM | 15,084 | 73,382 | 122 | N/A |
| | Book-Crossing | 110,739 | 442,746 | 52 | N/A |
| Node classification | ACM | 8,994 | 25,922 | 4 | 1,902 |
| | IMDB | 12,772 | 37,288 | 4 | 1,256 |

## C  Implementation details

All the models are randomly initialized and optimized using Adam [1] optimizers. For a fair comparison, the number of layers is set to two and the dimensionality of output node embeddings is the same across models. The node embedding $z$ for Last-FM has 16 dimensions and for the rest of the datasets

64 dimensions. Since datasets have a different number of samples, we train models for a different number of epochs; Last-FM (100), Book-Crossing (50), ACM (200), and IMDB (200). For link prediction, the neighborhood sampling algorithm [2] is used and the neighborhood size is 8 and 16 in Last-FM and Book-Crossing respectively. For node classification, the neighborhood size is 8 in all datasets. For the experiments on link prediction using **SELAR+Hint**, we train a learner network with attenuated weights, i.e., $\mathcal{V}_H(\xi_i^{(t,train)}; \Theta_H)^\gamma f^t(x_i^{(t,train)}; \mathbf{w}) + (1 - \mathcal{V}_H(\xi_i^{(t,train)}; \Theta_H)^\gamma) f_H^t(x_i^{(t,train)}; \mathbf{w})$, where $0 < \gamma \leq 1$. Hyperparameters such as learning rate and weight-decay rate are tuned using validation sets for all models. The test performance was reported with the best models on the validation sets. All the experiments use PyTorch [3] and the geometric deep learning extension library provided by Fey & Lenssen [4].

## D  Task selection results

Table 2: The average of the task-specific weighted loss on **Last-FM** and **Book-Crossing** datasets.

| Meta-paths (Last-FM) | Avg. | Meta-paths (Book-Crossing) | Avg. |
|---|---|---|---|
| user-item-actor-item | **7.675** | user-item* | **6.439** |
| user-item* | 7.608 | user-item-literary.series-item-user | 6.217 |
| user-item-appearing.in.film-item | 7.372 | item-genre-item | 6.163 |
| user-item-instruments-item | 7.049 | user-item-user-item | 6.126 |
| user-item-user-item | 6.878 | user-item-user | 6.066 |
| user-item-artist.origin-item | 6.727 | item-user-item | 6.025 |

∗ primary task

Our proposed methods identify useful auxiliary tasks and balance them with the primary task. In other words, the loss functions for tasks are differentially adjusted by the weighting function learned by SELAR+HintNet. To analyze the weights of the tasks, we calculate the average of the task-specific weighted loss. Table. 2 shows tasks in descending order of the task weights. 'user-item-actor-item' has the largest weight followed by 'user-item' (primary task), 'user-item-appearing.in.film-item', 'user-item-instruments-item', 'user-item-user-item' and 'user-item-artist.origin-item' on the Last-FM. It indicates that the preference of a given user is closely related to other items connected by an actor, e.g., specific edge type 'film.actor.film' in the knowledge graph. Moreover, our method focuses on 'user-item' interaction for the primary task. On the Book-Crossing data, our method has more attention to 'user-item' for the primary task and 'user-item-literary.series.item-user' which means that users who like a series book have similar preferences.

## E  Weighting function at different training stages

The weighting functions of our methods dynamically change over time. In Fig. 1, each row is the weighting function learned by SELAR+HintNet for GCN [5], GAT [6], GIN [7], and SGC [8] on Last-FM. From left, columns are from the first epoch, the epoch with the best validation performance, and the last epoch respectively. The positive and negative samples are illustrated in solid and dash lines respectively in Fig. 1. At the begging of training (the first epoch), one noticeable pattern is that the weighting function focuses more on 'easy' samples. At the epoch with the highest performance, easy samples are down-weighted and the weight is large when the loss is large. It implies that hard examples are more focused. At the last epoch, most weights converge to zero when the loss is extremely small or large in the last epoch. Since learning has almost completed, the weighting function becomes relatively smaller in most cases, e.g., GCN, GAT, and GIN. Especially, for GCN and GAT in the epoch with the highest performance, the weights are increasing and it means that our weighting function imposes that easy samples to smaller importance and more attention on hard samples. Among all tasks, the scale of weights in the primary task is relatively high compared to that of auxiliary tasks. This indicates that our method focuses more on the primary task.

Figure 1: Weightinf function $\mathcal{V}(\cdot)$ learnt by SELAR+Hint on Last-FM on GCN, GAT, GIN and SGC.

## Footnotes

*First two authors have equal contribution

†This work was done when the author worked at NAVER CLOVA.