[Reviews · NeurIPS 2020]

Review 1

Summary and Contributions: Proposes a graph neural network model with auxiliary learning.

Strengths: The overall quality is good. The problem is important. Experiments on node classification and link prediction are conducted.

Weaknesses: The novelty is incremental. Experiments could be improved.

Correctness: More illustration should be provided for the choice of auxiliary tasks.

Clarity: Well written paper.

Relation to Prior Work: More work should be compared.

Reproducibility: Yes

Additional Feedback: Detailed review. -- Summary The manuscript proposes SELAR, a graph neural network model with auxiliary learning. SELAR introduces auxiliary tasks (i.e., metapath prediction) to augment main task and learn better representations. A Hint network is further proposed for better optimization. Experiments on several datasets demonstrate that the proposed model outperforms some baseline methods for node classification and link prediction. Pros 1 The problem is important. 2 The overall quality of this paper is good. 3 A new graph neural network model with auxiliary learning is proposed. Experiments on node classification and link prediction are conducted. Cons/Questions 1 The novelty of the proposed model is incremental. The main contribution lies in introducing auxiliary tasks (meta-path prediction) to augment main task for better GNN. In addition, there could be many different auxiliary tasks and more reason/illustration should be provided for the choice of meta-path prediction. 2 Experiments could be improved. Since the authors consider heterogeneous graphs in this work, it is better to compare some graph neural network models for heterogeneous graphs such as: Graph Transformer Networks, NeurIPS 2019 Heterogeneous Graph Attention Networks, WWW 2019 Heterogeneous Graph Neural Network, KDD 2019 Heterogeneous Graph Transformer, WWW 2020 To summarize, the overall quality of this work is good while the novelty is incremental and the experiments could be improved.


Review 2

Summary and Contributions: 1. Propose a self-supervised learning method on a heterogeneous graph via meta-path prediction without additional data. 2. Automatically select meta paths (auxiliary tasks) to assist the primary task via meta-learning. 3. Design a Hint Network that helps the learner network to benefit from challenging auxiliary tasks. 4. Experiments built on various state-of-the-art GNNs show good results.

Strengths: 1. Idea of self-supervised learning method on graph learning is novel in this area. 2. The primary task is link prediction (or node classification) and auxiliary tasks are meta-path predictions to capture rich information of a heterogeneous graph. I think this is an interesting formulation of the problem. 3. Experiments built on various state-of-the art GNNs.

Weaknesses: 1. Can you give more explanation and comparisons with this very relevant paper? When Does Self-Supervision Help Graph Convolutional Networks? ICML2020 It seems some of the idea are very similar. 2. Do you have some ablative study for your components? Don’t know the performance contributions for each part, e.g. Hintnet. 3. Some of the performance in Table 1/2 seems to be very marginal. Can you explain the significance? 4. Have you try whether you have performance gain when the sample size is small or number of categories is small?

Correctness: I think it is OK

Clarity: Yes

Relation to Prior Work: Lack of some previous works

Reproducibility: Yes

Additional Feedback: =============== update =================== I have checked the rebuttal and other comments, and decided to maintain my rating. It would be interesting to see some theoretical insights on under what conditions auxiliary tasks are valuable, and when the method may fail.


Review 3

Summary and Contributions: This paper proposes a novel self-supervised auxiliary learning method using meta-paths to learn graph neural networks on heterogeneous graphs. The objective is to learn a primary task by predicting meta-paths as auxiliary tasks. The proposed method can identify an effective combination of auxiliary tasks and automatically balance them to improve the primary task. This method can be applied to any graph neural networks in a plug-in manner without manual labeling or additional data. (1) This paper proposes a self-supervised learning method on a heterogeneous graph via meta-path prediction without additional data. (2) The proposed framework automatically selects meta-paths (auxiliary tasks) to assist the primary task via meta-learning. (3) They develop Hint Network that helps the learner network to benefit from challenging auxiliary tasks. (4) The experiments show that meta-path prediction improves the representational power. (5) This is the first auxiliary task with meta-paths specifically designed for leveraging heterogeneous graph structure.

Strengths: (1) This paper is well written. (2) Pre-training GNNs with an auxiliary task is a important topic. They propose a pre-training strategy specifically designed for heterogeneous graphs. It is the first paper to do so. (3) The auxiliary task does not require additional labeled data. (4) They propose to use meta-learning technique to select the beneficial meta-paths to avoid negative transfer.

Weaknesses: (1) The related work is not very detailed. There are many pretraining methods on GNNs. For example, Strategies for Pre-training Graph Neural Networks, Weihua Hu, Bowen Liu, Joseph Gomes, Marinka Zitnik, Percy Liang, Vijay Pande, Jure Leskovec, ICLR, 2020. (2)The introduction of Hint Networks lacks of details. The motivation of hint network is unclear. The improvement of hint networks is limited on four datasets compared with SELAR. (3) There are four base GNNs in the experiments. Currently, when applying GNNs to large scale graphs, sampling partial graphs as input is a common strategy. For example, graphSAINT. It is nice to include those base GNNs.

Correctness: It seems correct to me.

Clarity: very clear

Relation to Prior Work: It is clearly discussed how this work differs from previous contributions.

Reproducibility: Yes

Additional Feedback:


Review 4

Summary and Contributions: This paper proposes a meta-learning framework (SELAR) to improve graph neural networks for heterogeneous graphs. Specifically, SELAR identifies a combination of auxiliary learning tasks from the training data by leveraging the unique and important structures (i.e., meta-paths) of heterogeneous graphs. These auxiliary tasks are self-supervised and they can improve the performance of a graph neural network on the primary task. Experiment results on node classification and link prediction tasks demonstrate that the proposed framework improves the performance of several existing graph neural networks on heterogeneous graphs.

Strengths: 1. This paper provides a novel way to automatically generate datasets for auxiliary learning in heterogeneous graphs without a manual labeling process, by utilizing the unique and important structure (i.e., meta-path) of heterogeneous graphs. 2. The proposed framework is very general and can serve as a plugin to improve the performance of any existing graph neural network on heterogeneous graphs. Hence, the reviewer thinks that it is a significant work since lots of existing models can be improved by this plugin. 3. This paper is sound in technical quality. The setups of the empirical studies are reasonable. The experiment results demonstrate the effectiveness of SELAR as a plugin to boost the performance of GNN, and verify the efficacy of each major component of this framework.

Weaknesses: 1. This paper does not apply the proposed framework to existing heterogeneous graph neural networks. According to the reviewers’ understanding, the proposed framework can be directly applied to existing heterogeneous GNNs, but it only shows the results of the framework on homogeneous GNNs. 2. Some important baselines are missing in the experiments. This paper only compares the performance of their framework with homogeneous graph neural networks, but it does not compare their method against existing heterogeneous GNNs (e.g., HAN) and heterogeneous graph embedding methods (e.g., metapath2vec). 3. Improvement in the node classification task is marginal. The ablation study on node classification seems to indicate the model benefits more from the meta-path auxiliary task itself rather than your framework.

Correctness: The methodology is derived correctly and explained clearly. The experimental setups are reasonable.

Clarity: This work is well written and easy to follow.

Relation to Prior Work: This paper clearly describes the differences between the proposed framework and existing auxiliary learning methods for graph neural networks. The related work also compares the proposed framework with auxiliary learning and meta-learning methods for other domains and applications. Overall, this paper is well-positioned. Most of the methodology and experiments of this work are described in detail. But some symbols used in the mathematical functions are not defined clearly, and there is no definitions (equations) given for the task-specific networks, HintNet, and the weighting functions \mathcal{V} and \mathcal{V}_{H}.

Reproducibility: Yes

Additional Feedback: 1) Line 89: according to the bolded characters, the abbreviation of your framework should be SELAL, not SELAR 2) Figure 3 caption: I believe the last equation should be written as -(1-p_t)^\gamma \log(\hat{y}) 3) Equation 2, 3, and 5 should only depend on \Theta, because w is determined by \Theta 4) Line 285: “more power” -> “more powerful” 5) Please add more or clearer explanations to define each non-trivial symbol used in the mathematical equations. E.g., what does x_i mean in Equation 3 and 4, is it the input feature of one node or input features of a pair of nodes, or is it task-specific? 6) You may give a brief introduction beforehand to some important terminologies about auxiliary learning / meta-learning. E.g., what is meta-dataset? 7) It would be better if dataset statistics are included in the paper. 8) It is not clear whether you have repeated the experiments for several repetitions or just run the experiments for once. 9) What is the difference between auxiliary learning and meta-learning?

[Author Response · NeurIPS 2020]

We thank all four reviewers for unanimous support for the paper and constructive comments. To recap, our submission is the first paper to study meta-path prediction as an auxiliary task to train GNNs on heterogeneous graphs. Our meta learning-based frameworks adaptively balance auxiliary tasks (meta-path prediction) with the primary task (link prediction or node classification). The proposed methods improve representational power without any additional data or labels. Overall, reviewers are positive about our contributions: [R5] "The proposed framework is very general and can serve as a plugin to improve the performance of any existing GNNs.", [R4] "They propose a pre-training strategy specifically designed for heterogeneous graphs. It is the 'first' paper to do so.", and [R1] "The problem is important. The overall quality of this paper is good." Also, we will address questions raised by the reviewers below hoping for more vigorous support.

**Q1 [R1, R2, R5]. Is the proposed method applicable to existing heterogeneous GNNs such as GTNs [1]?**
Yes, as [R4] pointed out, our framework can be applied to any GNNs in a plug-in manner. Table. 1 shows our framework consistently improves the representational power of Graph Transformer Networks (GTNs) for heterogeneous graphs in both link prediction (Last-FM) and node classification (IMDB). We have repeated the experiments three times as requested by **[R5]**. Also, we added the HINT column to evaluate the efficacy of HintNet itself as suggested by **[R2]**.

Table 1: Performance of GTN trained by various learning schemes.

| Dataset | Vanilla | w/o meta-path | Ours | | |
| | | | w/ meta-path | Hint | SELAR | SELAR+Hint |
|---|---|---|---|---|---|---|
| Last-FM | $0.7836\pm0.0030$ | $0.7744\pm0.0022$ | $0.7865\pm0.0042$ | $0.7883\pm0.0054$ | $0.7978\pm0.0018$ | **$0.8053$**$\pm0.0064$ |
| IMDB | $0.5804\pm0.0073$ | $0.5792\pm0.0017$ | $0.5818\pm0.0088$ | $0.5889\pm0.0112$ | $0.5994\pm0.0097$ | **$0.6063$**$\pm0.0046$ |

**Q2 [R4, R5]. Performance gain from the Hint Network.**
Table. 1 in our supplementary materials show clearer improvement by the Hint Network rather than the results in the main paper. Our Hint Network with a regularizer improves the performance of GNNs along with SELAR in most cases. In 7 out of 8 cases, SELAR+HintNet outperforms SELAR. We found that imposing a regularizer on HintNet is helpful. We will move these results to the main paper in our final version.

**Q3 [R4, R5]. Motivation and details of Hint Network.**
We introduced Hint Networks in Section 3.3 in the main paper. Compared to primary tasks such as link prediction and node classification, the meta-path prediction might be more difficult to be learned. For instance, when training a GNN on a large graph, some important intermediate nodes and edges may not be available in a small mini-batch. Moreover, GNNs with a small number of layers are not capable to learn long meta-paths. Our Hint Network makes the challenging tasks more solvable by correcting the answer at learner's need with hub nodes which have rich connections with other nodes. Roughly speaking, meta-learning for SELAR allows GNNs to learn which task to learn (meta-path selection) and meta-learning for HintNet adjusts the difficulty level of a given task. In our experiments in the main paper, HintNet $f_H^t$ is the exactly same model as learner $f^t$ but it uses the augmented mini-batch with hub nodes. Our method can learn from either original topology or augmented topology with hub nodes at learner's needs.

**Q4 [R1, R2, R4]. Why meta-path prediction? More discussion about self-supervised learning on graphs.**
Great question! We found that meta-path prediction is an effective self-supervision task to train GNNs on heterogeneous graphs. Our experiments support that meta-path prediction itself improves the representational power of GNNs significantly. As reviewers pointed out, self-supervision and pre-training of GNNs have recently been studied in the literature [2, 3]. Weihua Hu et al. [2] have introduced effective strategies for pre-training GNNs such as attribute masking and context prediction. Separated from the pre-training and fine-tuning strategy, [3] studies multi-task learning and analyzes why the pretext tasks are useful for GCNs and its close variants. However, one problem with both pre-training and multi-task learning strategies is that all the auxiliary tasks are not beneficial for the downstream applications. Motivated by this problem, we studied 'auxiliary learning' for GNNs that explicitly focuses on the primary task. Auxiliary learning is similar to multi-task learning but it is not obliged to improve the performance of any auxiliary tasks. The auxiliary tasks just assist the primary task. We proposed frameworks that utilize multiple auxiliary tasks without harming the primary task. This can be formulated as a general meta-learning problem and our methods can incorporate all the self-supervision tasks mentioned above as auxiliary tasks. Also, we believe that the same idea can be used in any representation learning. We will add more discussion about related work in the final version if accepted.

# References

[1] Seongjun Yun et al. Graph transformer networks. In *NeurIPS*, pages 11983–11993, 2019.
[2] Weihua Hu et al. Strategies for pre-training graph neural networks. In *ICLR*, 2020.
[3] Yuning You et al. When does self-supervision help graph convolutional networks? *ICML*, 2020.


[Meta-Review · NeurIPS 2020]

Overall, this paper was borderline. The main strengths of the paper is that it's addressing an important problem, and that the experiments and presentation of the work are both solid. The reviewers appreciated the new experiments in the author response. The main weaknesses of the paper are the novelty of the overall idea and the increased efforts required for customized model design for different datasets/graphs. In the balance, I think that the merits outweigh the flaws.